# First-Trimester Screening for HELLP Syndrome—Prediction Model Based on MicroRNA Biomarkers and Maternal Clinical Characteristics

**DOI:** 10.3390/ijms24065177

**Published:** 2023-03-08

**Authors:** Ilona Hromadnikova, Katerina Kotlabova, Ladislav Krofta

**Affiliations:** 1Department of Molecular Biology and Cell Pathology, Third Faculty of Medicine, Charles University, 100 00 Prague, Czech Republic; 2Institute for the Care of the Mother and Child, Third Faculty of Medicine, Charles University, 147 00 Prague, Czech Republic

**Keywords:** cardiovascular diseases, first-trimester screening, gene expression, HELLP syndrome, microRNAs, prediction, whole peripheral venous blood

## Abstract

We evaluated the potential of cardiovascular-disease-associated microRNAs for early prediction of HELLP (hemolysis, elevated liver enzymes, and low platelets) syndrome. Gene expression profiling of 29 microRNAs was performed on whole peripheral venous blood samples collected between 10 and 13 weeks of gestation using real-time RT-PCR. The retrospective study involved singleton pregnancies of Caucasian descent only diagnosed with HELLP syndrome (*n* = 14) and 80 normal-term pregnancies. Upregulation of six microRNAs (miR-1-3p, miR-17-5p, miR-143-3p, miR-146a-5p, miR-181a-5p, and miR-499a-5p) was observed in pregnancies destined to develop HELLP syndrome. The combination of all six microRNAs showed a relatively high accuracy for the early identification of pregnancies destined to develop HELLP syndrome (AUC 0.903, *p* < 0.001, 78.57% sensitivity, 93.75% specificity, cut-off > 0.1622). It revealed 78.57% of HELLP pregnancies at a 10.0% false-positive rate (FPR). The predictive model for HELLP syndrome based on whole peripheral venous blood microRNA biomarkers was further extended to maternal clinical characteristics, most of which were identified as risk factors for the development of HELLP syndrome (maternal age and BMI values at early stages of gestation, the presence of any kind of autoimmune disease, the necessity to undergo an infertility treatment by assisted reproductive technology, a history of HELLP syndrome and/or pre-eclampsia in a previous gestation, and the presence of trombophilic gene mutations). Then, 85.71% of cases were identified at a 10.0% FPR. When another clinical variable (the positivity of the first-trimester screening for pre-eclampsia and/or fetal growth restriction by the Fetal Medicine Foundation algorithm) was implemented in the HELLP prediction model, the predictive power was increased further to 92.86% at a 10.0% FPR. The model based on the combination of selected cardiovascular-disease-associated microRNAs and maternal clinical characteristics has a very high predictive potential for HELLP syndrome and may be implemented in routine first-trimester screening programs.

## 1. Introduction

HELLP syndrome (hemolysis, elevated liver enzymes, and low platelets) is a very rare pregnancy-related disorder with a general prevalence ranging from 0.2% to 0.9% [1,2,3,4,5,6,7,8]. It occurs separately or in an association with severe pre-eclampsia (PE), where the incidence increases to 4–24% [2,6,9,10,11,12,13].

High maternal and fetal morbidity and mortality were reported due to the appearance of severe maternal and neonatal complications [1,9,10,14,15,16,17,18,19,20,21,22,23,24,25,26].

The diagnosis and the severity of HELLP syndrome are evaluated using two classification systems: the Tennessee classification and the Mississippi classification [20,27,28,29,30,31].

Both classification systems are based on the number of platelets in the blood (PLT), the serum levels of aspartate (AST) or alanine (ALT) aminotransferases and lactate dehydrogenase (LDH), and signs of hemolysis in a peripheral blood smear [20,27,28,29,30,31]. 

In the Tennessee classification, the platelet count is usually below 100 × 10^9^/L, serum AST levels are usually above 70 IU/L, and serum LDH levels are usually above 600 IU/L. While complete (full) HELLP syndrome requires the fulfilment of all diagnostic criteria, in incomplete (partial) HELLP syndrome only one or two of the abovementioned criteria are diagnosed [27,28,29]. 

In the Mississippi classification, the most severe HELLP syndrome (also termed class 1) is represented by a platelet count below 50 × 10^9^/L, moderate HELLP syndrome (class 2) by a blood platelet number in the range of 50 × 10^9^/L to 100 × 10^9^/L, and the mild form of HELLP syndrome (class 3) by a platelet count between 100 × 10^9^/L and 150 × 10^9^/L. Class 1 and class 2 HELLP syndrome are further defined by serum AST or ALT levels above 70 IU/L, while class 3 HELLP syndrome is defined by levels above 40 IU/L. In all cases, serum LDH levels are above 600 IU/L [20,30,31].

Several risk factors predisposing to the development of HELLP syndrome have already been identified. These factors are Caucasian ethnicity, nulliparity, history of gestational hypertensive disorders or HELLP syndrome in a previous gestation, multiple pregnancy at an advanced maternal age, and increased levels of mean arterial pressure (MAP) assessed in first-trimester screening [32,33,34,35,36,37].

The pathophysiology of HELLP syndrome has not yet been fully discovered. However, the activation of endothelial cells, increased production of antiangiogenic factors and microvascular injury resulting in microangiopathic anemia, periportal hepatic necrosis, and thrombocytopenia are probably induced by the ischemia occurring in placental tissue [8,37,38,39,40,41,42,43,44,45,46].

Up to now, only a few models for the prediction of HELLP syndrome have been developed.

A logistic regression model (LRM) based on racial origin, nulliparity, history of HELLP syndrome, and PE showed an area under the curve (AUC) of 0.800, 75.0% sensitivity, and 79.0% specificity. The detection rate, with a 10.0% false-positive rate (FPR), reached 55.0% of cases [36].

Recently, a neuro-fuzzy model for the identification and prediction of HELLP syndrome has been developed. This novel model reached an AUC of 0.829 and a precision level of 0.685, but only seven pregnant women with HELLP syndrome were included in the study [47]. Unfortunately, the maternal clinical variables involved in this model were not stated.

The aim of our study was to develop an efficient early predictive model for HELLP syndrome that could be implemented in the current algorithms of first-trimester screening. Since HELLP syndrome usually develops in the third trimester of gestation, the availability of an early predictive model for HELLP syndrome is desirable. 

Initially, we focused on the identification of risk factors associated with the later development of HELLP syndrome occurring separately or accompanying severe PE.

Afterwards, we were interested to see whether there was any other potential way to improve the detection rate of our novel HELLP predictive model based on maternal clinical characteristics only. 

Recently, we reported an altered expression profile of microRNAs associated with the cardiovascular system in pregnant women affected with chronic hypertension [48] and in normotensive early pregnancies with subsequent onset of gestational hypertension (GH) [48], PE [48], fetal growth restriction (FGR) [49], small for gestational age (SGA) [49], preterm delivery [50], and/or gestational diabetes mellitus (GDM) [51]. Therefore, we were interested to determine whether an altered expression profile of microRNAs associated with the cardiovascular system might also be present in pregnancies developing HELLP syndrome.

We performed at early gestational stages whole peripheral blood gene expression profiling of 29 selected microRNAs demonstrated previously to play a crucial role in the development and maintenance of homeostasis in the cardiovascular system and in the pathophysiology of cardiovascular and cerebrovascular diseases (Table 1) [51].

Finally, we examined the reliability of our novel early predictive model for HELLP syndrome based on the cardiovascular-disease-associated microRNA biomarkers combined with maternal clinical characteristics identified in our study as independent risk factors for the development of HELLP syndrome.

## 2. Results

### 2.1. Identification of Risk Factors for the Development of HELLP Syndrome

The clinical characteristics of cases (HELLP-syndrome pregnancies) and controls (normal-term pregnancies) are outlined in Table 2.

The following independent risk factors for the development of HELLP syndrome with or without PE were identified at early gestational stages: the presence of any autoimmune disease, an infertility treatment by assisted reproductive technologies, the occurrence of HELLP syndrome and/or PE in a previous gestation, the presence of mutations in trombophilic genes, and the positivity of first-trimester PE/FGR screening by the Fetal Medicine Foundation (FMF) algorithm.

### 2.2. Altered Expression Profiles of MicroRNAs during the First Trimester of Gestation in Pregnancies Developing HELLP Syndrome

Whole peripheral blood first-trimester expression profiles of microRNAs were compared between pregnancies that developed HELLP-syndrome and normal-term pregnancies. 

Increased levels of miR-1-3p (*p* < 0.001), miR-17-5p (*p* = 0.010), miR-143-3p (*p* = 0.011), miR-146a-5p (*p* < 0.001), miR-181a-5p (*p* = 0.001), and miR-499a-5p (*p* < 0.001) were detected during the first trimester of gestation in pregnancies developing HELLP syndrome (Figure 1).

Individual microRNAs differentiated between normal-term pregnancies and pregnancies developing HELLP syndrome with various sensitivities at a 10.0% FPR. MiR-1-3p (64.29%) and miR-499a-5p (50.0%) showed the best sensitivities, miR-146a-5p (35.71%) and miR-181a-5p (28.57%) showed moderate sensitivities, and miR-17-5p (14.29%) and miR-143-3p (14.29%) showed the lowest sensitivities (Figure 1). 

### 2.3. The Prediction Model for HELLP Syndrome—The Combination of Six MicroRNAs Only

The prediction model for HELLP syndrome based on the combination of six microRNAs only (miR-1-3p, miR-17a-5p, miR-143a-3p, miR-146a-5p, miR-181a-5p, and miR-499a-5p) identified pregnancies developing HELLP syndrome with relatively high accuracy (AUC 0.903, *p* < 0.001, 78.57% sensitivity, 93.75% specificity, cut-off > 0.1622). A total of 78.57% of pregnancies destined to develop HELLP syndrome was revealed at early stages of gestation with a 10.0% FPR (Figure 2).

### 2.4. The Prediction Model for HELLP Syndrome Based on Selected Maternal Clinical Characteristics Only

The prediction model for HELLP syndrome based on the combination of six selected maternal clinical characteristics only (maternal age and BMI values at early gestational stages, the presence of any autoimmune disease, an infertility treatment by assisted reproductive technologies, the occurrence of HELLP syndrome and/or PE in a previous gestation, and the presence of mutations in trombophilic genes) identified pregnancies developing HELLP syndrome with a relatively high accuracy (AUC 0.862, *p* < 0.001, 71.43% sensitivity, 97.50% specificity, cut-off >0.1244). A total of 71.43% of pregnancies destined to develop HELLP syndrome was revealed at early gestational stages with a 10.0% FPR. The addition of another maternal clinical characteristic to the prediction model (the positivity of first-trimester PE/FGR screening by FMF) also revealed 71.43% of HELLP pregnancies with a 10.0% FPR (AUC 0.849, *p* < 0.001, 71.43% sensitivity, 97.50% specificity, cut-off > 0.1406) (Figure 3).

### 2.5. The Full Prediction Model for HELLP Syndrome Based on the Combination of Six MicroRNAs and Selected Maternal Clinical Characteristics

The full prediction model for HELLP syndrome based on the combination of six microRNAs (miR-1-3p, miR-17a-5p, miR-143a-3p, miR-146a-5p, miR-181a-5p, and miR-499a-5p) and six selected maternal clinical characteristics (maternal age and BMI values at early gestational stages, the presence of any autoimmune disease, an infertility treatment by assisted reproductive technologies, the occurrence of HELLP syndrome and/or PE in a previous gestation, and the presence of mutations in trombophilic genes) identified pregnancies developing HELLP syndrome with relatively high accuracy (AUC 0.979, *p* < 0.001, 100.0% sensitivity, 86.25% specificity, cut-off > 0.0494). A total of 85.71% of pregnancies destined to develop HELLP syndrome was revealed at early gestational stages with a 10.0% FPR. The addition of another maternal clinical characteristic to the prediction model (the positivity of first-trimester PE/FGR screening by FMF) revealed 92.86% of HELLP pregnancies at a 10.0% FPR (AUC 0.975, *p* < 0.001, 92.86% sensitivity, 92.50% specificity, cut-off > 0.1110) (Figure 4). 

## 3. Discussion

Initially, we focused on the identification of risk factors associated with later development of HELLP syndrome occurring separately or accompanying severe PE.

We identified an increased incidence of HELLP syndrome in patients with already diagnosed autoimmune diseases, such as SLE, APS, SS, RA, T1DM, and coeliac disease, which has not yet been reported.

Furthermore, we observed a higher incidence of HELLP syndrome in patients undergoing an infertility treatment by assisted reproductive technology (ART), which was also reported as an independent risk factor for the onset of hypertensive disorders during pregnancy, such as GH or PE [52,53,54] and FGR [55].

Qin et al. [56] pointed to the fact that singleton pregnancies undergoing assisted reproductive technologies are at a higher risk of adverse outcomes and recommended that they be managed as high-risk pregnancies. The ART singleton pregnancies had a significant risk of pregnancy-induced hypertension, GDM, placenta previa, placental abruption, antepartum hemorrhage, postpartum hemorrhage, polyhydramnios, oligohydramnios, cesarean sections, preterm birth, small for gestational age, perinatal mortality, and congenital malformation [56].

Similarly to other researchers [32,33,34,35,36,37], we confirmed that a history of HELLP syndrome and/or PE in a previous gestation represents a risk factor predisposing to the development of HELLP syndrome. 

Furthermore, we demonstrated that the presence of trombophilic gene mutations is more frequent in pregnancies developing HELLP syndrome. This finding is congruent with the observations of Muetze et al. [39], who reported that mutation in factor V Leiden is associated with HELLP syndrome in women of Caucasian descent.

Unsurprisingly, we also detected a higher incidence of HELLP syndrome in patients with positive first-trimester PE/FGR screening by FMF [57,58,59,60].

All of the maternal risk factors identified by our research group were placed in a model together with maternal age and BMI values at early gestational stages to assess their common predictive potential for later development of HELLP syndrome. This prediction model for HELLP syndrome identified pregnancies developing HELLP syndrome with relatively high accuracy, since it was able to reveal 71.43% of cases with a 10.0% FPR. The addition of another maternal clinical characteristic to the prediction model (the positivity of first-trimester PE/FGR screening by FMF) did not yield a better detection rate, since it was able to detect the same proportion of pregnancies with HELLP syndrome (71.43% of cases with a 10.0% FPR).

Our model based on maternal risk factors only showed better performance than the logistic regression model demonstrated previously by Oliveira et al. [36]. This model was based on racial origin, nulliparity, and the occurrence of HELLP syndrome and PE in a previous gestation, and reached a detection rate of 55.0% of cases only, with a 10.0% FPR.

A recently developed neuro-fuzzy model for HELLP syndrome identification and prediction [47] also showed a lower discrimination power than our novel model based on six or seven basic maternal clinical characteristics, including maternal age and BMI values at early gestational stages, the presence of any autoimmune disease, an infertility treatment by assisted reproductive technologies, the occurrence of HELLP syndrome and/or PE in a previous gestation, and the presence of mutations in trombophilic genes, or the positivity of first-trimester PE/FGR screening by FMF [57,58,59,60]. 

Afterwards, we were interested to see whether there was any other potential way to improve the detection rate of our novel HELLP predictive model based on maternal clinical characteristics only. Therefore, we evaluated the predictive potential of microRNAs that play a crucial role in the development and maintenance of homeostasis in the cardiovascular system and in the pathophysiology of cardiovascular and cerebrovascular diseases (Table 1) [51].

Gene expression of preselected microRNAs associated with the cardiovascular system was retrospectively studied in peripheral blood during the first trimester in pregnancies subsequently developing HELLP syndrome and in normal-term pregnancies selected as a matched control group based on the equality of the period of biological sample storage and gestational age at sampling.

Currently, the upregulation of six microRNAs associated with the cardiovascular system (miR-1-3p, miR-17-5p, miR-143-3p, miR-146a-5p, miR-181a-5p, and miR-499a-5p) was observed during the early gestational stages in pregnancies developing HELLP syndrome. The combination of these six microRNA biomarkers only identified pregnancies developing HELLP syndrome with relatively high accuracy. A total of 78.57% of pregnancies destined to develop HELLP syndrome was revealed at early gestational stages with a 10.0% FPR. This is an optimistic result if we take into consideration that any knowledge of maternal clinical characteristics is required to achieve such a high discrimination power. Nevertheless, the availability of early screening for HELLP syndrome in clinical practice depends on the successful validation of microRNA biomarkers in consecutive prospective cohort studies and the acquisition of CE and IVDR certifications. The advantage of such a screening based on microRNA biomarkers only is that additional information about maternal characteristics is not needed.

Interestingly, apart from HELLP-syndrome pregnancies, miR-1-3p also showed an altered early expression profile in pregnancies destined to develop SGA [49] and GDM [51]. MiR-143-3p displayed an aberrant expression profile in pregnancies developing PE [48]. Besides HELLP syndrome, an altered early expression profile of miR-146a-5p appeared in pregnancies developing PE [48] and FGR or SGA [49]. An aberrant early expression profile of miR-181a-5p was present in most pregnancies regardless of the type of pregnancy-related complication: GH or PE [48], FGR or SGA [49], GDM [51], or HELLP syndrome. MiR-499a-5p expression profiles were also observed to be dysregulated in early stages of gestation in pregnancies destined to develop GDM [51].

When these six cardiovascular-disease-associated microRNAs were added to the predictive model based on six maternal clinical characteristics identified by our research group as risk factors for the onset of HELLP syndrome, the discrimination power increased to 85.71% with a 10.0% FPR. The addition of another maternal clinical characteristic to the prediction model (the positivity of first-trimester PE/FGR screening by FMF) did not increase the AUC but significantly increased the sensitivity with a 10.0% FPR. Using this approach, 92.86% of HELLP pregnancies were finally detected.

To our knowledge, no studies on the early prediction of HELLP syndrome during the first trimester through screening of extracellular microRNAs in maternal body fluids (plasma/serum) or peripheral blood samples are currently available.

Just one study describing the identification of differentially expressed microRNAs in serum samples of patients with clinical manifestations of HELLP syndrome is available. Upregulation of miR-122, miR-758, and miR-133a was detected in a group of patients with HELLP syndrome [61]. Concerning miR-122 and miR-758, we did not examine their expression levels in maternal peripheral venous blood leukocytes. In addition, we studied miR-133a-3p, whose expression levels did not differ in early stages of gestation between pregnancies developing HELLP syndrome and those with normal courses of gestation delivering at term.

Recently, bioinformatics analysis of microarray data has identified hub genes (KIT, JAK2, LEP, EP300, HIST1H4L, HIST1H4F, HIST1H4H, MMP9, THBS2, and ADAMTS) as diagnostic biomarkers of HELLP syndrome. MiR-34a-5p was demonstrated to be most associated with hub genes [46]. Unfortunately, the expression profile of miR-34a-5p was not studied in our group of patients. 

## 4. Materials and Methods

### 4.1. Patient Cohort

The peripheral blood sampling was performed within the framework of the first-trimester prenatal screening between 10 and 13 gestational weeks within the period November 2012–May 2018. In total, 4187 samples were collected from Caucasian singleton pregnancies. In the end, 3028 pregnant women delivered on site. In all, 14 of 3028 pregnant women were diagnosed with HELLP syndrome. 

The diagnosis and the severity of HELLP syndrome were assessed using two classification systems: the Tennessee classification and the Mississippi classification [20,27,28,29,30,31].

Complete (full) HELLP syndrome developed in 3 cases and incomplete (partial) HELLP syndrome was diagnosed in 11 cases.

The most severe form of HELLP syndrome (also termed class 1) was present in 3 cases, the moderate form (class 2) in 4 cases, and the mild form (class 3) in 7 cases. 

Seven pregnancies were diagnosed with HELLP syndrome only, and seven pregnancies had HELLP syndrome associated with severe PE.

Clinical management guidelines issued by the American College of Obstetricians and Gynecologists (ACOG) including diagnostic criteria for pre-eclampsia were followed [62].

The selection of controls was performed with respect to the equality of gestational age at the time of sample collection and the period of storage of biological samples. The control group consisted of 80 normal-term pregnancies. The control group delivered healthy newborns after 37 gestational weeks with weights over 2500 g.

### 4.2. Processing of Samples

Processing of samples, reverse transcription (RT), and real-time qPCR analyses were performed as described previously [48,49,50,51].

In detail, leukocyte lysates were prepared from 200 µL peripheral blood samples using the QIAamp RNA Blood Mini Kit (Qiagen, Hilden, Germany) and were stored in a mixture of RLT buffer and β-mercaptoethanol (β-ME) at −80 °C. 

The MirVana microRNA Isolation kit (Ambion, Austin, USA) was used to isolate RNA fractions highly enriched for small RNAs. 

Gene expression of microRNAs associated with the cardiovascular system (miR-1-3p, miR-16-5p, miR-17-5p, miR-20a-5p, miR-20b-5p, miR-21-5p, miR-23a-3p, miR-24-3p, miR-26a-5p, miR-29a-3p, miR-92a-3p, miR-100-5p, miR-103a-3p, miR-125b-5p, miR-126-3p, miR-130b-3p, miR-133a-3p, miR-143-3p, miR-145-5p, miR-146-5p, miR-155-5p, miR-181a-5p, miR-195-5p, miR-199a-5p, miR-210-3p, miR-221-3p, miR-342-3p, miR-499a-5p, and miR-574-3p) was studied.

RT and real-time qPCR analyses were performed via TaqMan MicroRNA Assays (Applied Biosystems, Branchburg, NJ, USA) on a 7500 Real-Time PCR System under standard TaqMan PCR conditions.

The relative expression of microRNA genes was assessed using the delta-delta Ct method [63]. The endogenous controls (RNU58A and RNU38B) were used to normalize microRNA gene expression data [64,65]. 

### 4.3. Statistical Analysis

MicroRNA gene expression was compared between cases and controls using the Mann–Whitney test. 

Box plots displaying the medians, 75th and 25th percentiles, outliers (circles), and extremes (asterisks) were produced using Statistica software (version 9.0; StatSoft, Inc., Tulsa, OK, USA).

Receiver operating characteristic (ROC) curves displayed the areas under the curves (AUCs), cut-off-point-associated sensitivities, specificities, positive and negative likelihood ratios (LR+, LR−), and sensitivities at a 10.0% false-positive rate (FPR) (MedCalc Software bvba, Ostend, Belgium). 

To select the best microRNA combinations, logistic regression with subsequent ROC curve analyses was applied (MedCalc Software bvba, Ostend, Belgium). This statistical approach was also applied to develop novel early predictive models for HELLP syndrome based on the combination of appropriate microRNAs only, maternal clinical characteristics only, and the combination of appropriate microRNAs and maternal clinical characteristics (MedCalc Software bvba, Ostend, Belgium).

## 5. Conclusions

Consecutive, large-scale retrospective analyses have to be performed to verify the reliability of our novel early predictive model for HELLP syndrome based on the combination of microRNAs associated with the cardiovascular system (miR-1-3p, miR-17a-5p, miR-143a-3p, miR-146a-5p, miR-181a-5p, and miR-499a-5p) and maternal clinical characteristics (maternal age and BMI values at early gestational stages, the presence of any autoimmune disease, an infertility treatment by assisted reproductive technologies, the occurrence of HELLP syndrome and/or PE in a previous gestation, the presence of mutations in trombophilic genes, and, alternatively, the positivity of first-trimester PE/FGR screening by FMF). 

The model based on the combination of six cardiovascular-disease-associated microRNA biomarkers and six maternal clinical characteristics has a very high discrimination power. It is able to detect 85.71% of cases with a 10.0% FPR. The addition of another maternal clinical characteristic to this particular prediction model (the positivity of first-trimester PE/FGR screening by FMF) may significantly increase the detection rate to 92.86% of cases. 

In addition, consecutive prospective cohort studies are needed to validate the suggested early predictive model for HELLP syndrome. The availability of early screening for HELLP syndrome in clinical practice depends on the successful validation of microRNA biomarkers in consecutive prospective cohort studies and the acquisition of CE and IVDR certifications. Only high-risk pregnancies identified firstly by the early predictive model based on maternal characteristics may be further screened using microRNA biomarkers to achieve a reasonable cost–benefit ratio for the early predictive model for HELLP syndrome.

## 6. Patents

National patent application, Industrial Property Office, Czech Republic (patent No. PV 2022-505).

## Figures and Tables

**Figure 1 ijms-24-05177-f001:**
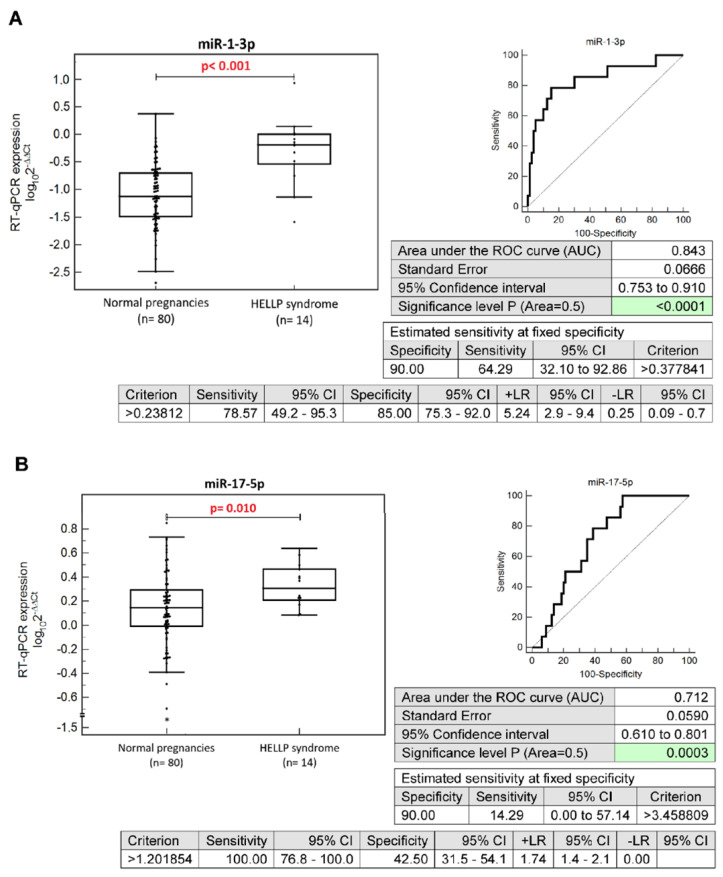
Differentiation between patients with HELLP-syndrome and normal-term pregnancies at early gestational stages based on peripheral blood microRNA expression profiles. Up-regulation of miR-1-3p (**A**), miR-17-5p (**B**), miR-143-3p (**C**), miR-146a-5p (**D**), miR-181a-5p (**E**), and miR-499a-5p (**F**) was observed. CI, confidence interval; +LR, positive likelihood ratio; −LR, negative likelihood ratio.

**Figure 2 ijms-24-05177-f002:**
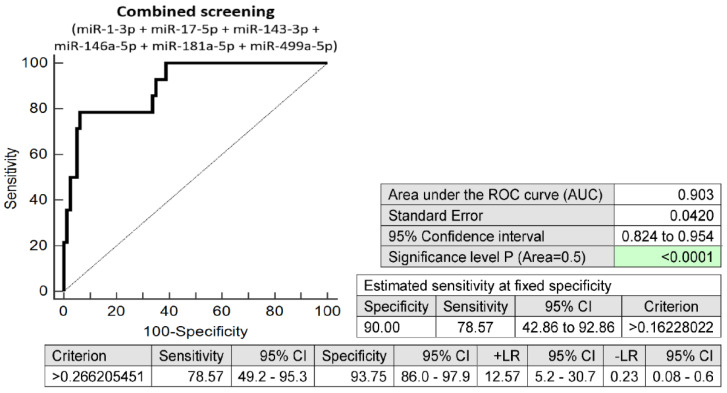
ROC analysis: the combination of six microRNAs only (miR-1-3p, miR-17a-5p, miR-143a-3p, miR-146a-5p, miR-181a-5p, and miR-499a-5p). With a 10.0% FPR, 78.57% of pregnancies developing HELLP syndrome were correctly identified. CI, confidence interval; +LR, positive likelihood ratio; -LR, negative likelihood ratio; FPR, false-positive rate.

**Figure 3 ijms-24-05177-f003:**
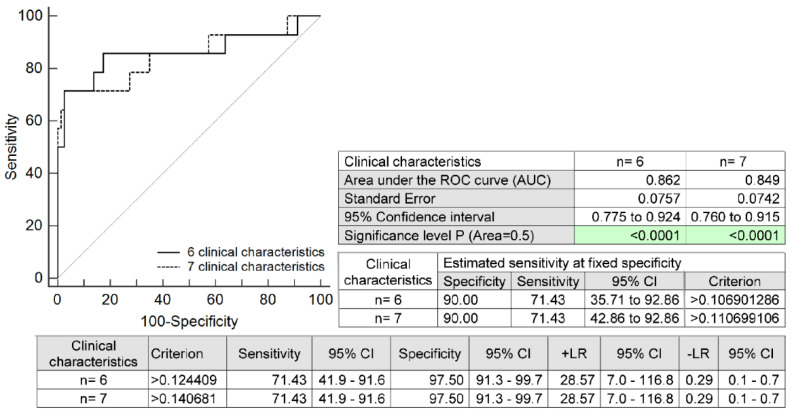
ROC analysis: the combination of six maternal clinical characteristics (maternal age and BMI values at early gestational stages, the presence of any autoimmune disease, an infertility treatment by assisted reproductive technologies, the occurrence of HELLP syndrome and/or PE in a previous gestation, and the presence of mutations in trombophilic genes). In the case of the combination of seven maternal clinical characteristics, the positivity of first-trimester PE/FGR screening by FMF was added. With a 10.0% FPR, 71.43% of pregnancies destined to develop HELLP syndrome were correctly identified. CI, confidence interval; +LR, positive likelihood ratio; −LR, negative likelihood ratio; BMI, body mass index; PE, pre-eclampsia; FGR, fetal growth restriction; FMF, Fetal Medicine Foundation; FPR, false-positive rate.

**Figure 4 ijms-24-05177-f004:**
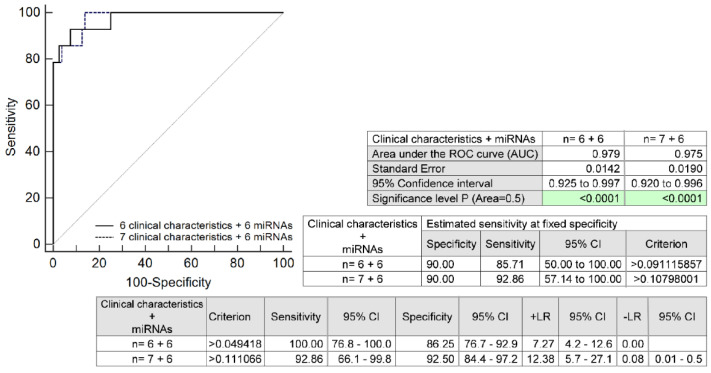
ROC analysis: the combination of six microRNAs (miR-1-3p, miR-17a-5p, miR-143a-3p, miR-146a-5p, miR-181a-5p, and miR-499a-5p) and six maternal clinical characteristics (maternal age and BMI values at early gestational stages, the presence of any autoimmune disease, an infertility treatment by assisted reproductive technologies, the occurrence of HELLP syndrome and/or PE in a previous gestation, and the presence of mutations in trombophilic genes). In the case of the combination of six microRNA biomarkers and seven maternal clinical characteristics, the positivity of first-trimester PE/FGR screening by FMF was added. With a 10.0% FPR, 85.71% and 92.86% of pregnancies developing HELLP syndrome were correctly identified. CI, confidence interval; +LR, positive likelihood ratio; -LR, negative likelihood ratio; BMI, body mass index; PE, pre-eclampsia; FGR, fetal growth restriction; FMF, Fetal Medicine Foundation; FPR, false-positive rate.

**Table 1 ijms-24-05177-t001:** The role of microRNAs in the development and maintenance of homeostasis in the cardiovascular system and in the pathophysiology of cardiovascular and cerebrovascular diseases.

	miR-1-3p	miR-16-5p	miR-17-5p	miR-20a-5p	miR-20b-5p	miR-21-5p	miR-23a-3p	miR-24-3p	miR-26a-5p	miR-29a-3p	miR-92a-3p	miR-100-5p	miR-103a-3p	miR-125b-5p	miR-126-3p	miR-130b-3p	miR-133a-3p	miR-143-3p	miR-145-5p	miR-146a-5p	miR-155-5p	miR-181a-5p	miR-195-5p	miR-199a-5p	miR-210-3p	miR-221-3p	miR-342-3p	miR-499a-5p	miR-574-3p
Homeostasis of the cardiovascular system						+																							
Angiogenesis																				+									
Cardiac development			+																										
Cardiac regeneration																												+	
Adipogenic differentiation			+																										
Obesity							+						+								+	+				+	+		
Insulin resistance					+	+							+																
Gestational diabetes mellitus		+	+	+						+				+	+	+			+		+		+	+			+		
Diabetes mellitus (T1DM, T2DM) and its complications	+	+	+	+	+	+		+	+	+		+	+	+	+	+	+		+	+	+	+	+	+			+		+
Hypercholesterolemia								+								+					+								
Metabolic syndrome																						+							
Hypertension						+						+	+						+		+	+		+					
Atherosclerosis											+									+	+	+			+	+			
Myocardial infarction	+	+	+			+			+	+	+		+	+	+			+	+	+		+			+	+		+	+
Cerebral ischemic events		+	+				+							+		+		+	+	+		+		+		+		+	+
Coronary heart disease		+	+				+	+	+	+	+		+				+	+		+	+	+	+	+	+	+	+		+
Pulmonary hypertension				+					+	+			+					+						+		+			
Heart failure		+			+	+	+		+	+	+	+					+						+	+		+			

T1DM: type 1 diabetes mellitus; T2DM: type 2 diabetes mellitus.

**Table 2 ijms-24-05177-t002:** Clinical characteristics of the study and control groups.

	Normal-Term Pregnancies (*n* = 80)	HELLP Overall (*n* = 14)	HELLP without PE (*n* = 7)	HELLP with PE (*n* = 7)	*p*-Value ^1^ Odds Ratio (OR) (95%CI)	*p*-Value ^2^ Odds Ratio (OR) (95%CI)	*p*-Value ^3^ Odds Ratio (OR) (95%CI)
*Maternal characteristics*							
Chronic hypertension	0 (0%)	0 (0%)	0 (0%)	0 (0%)	0.396 OR: 5.552 (0.106–291.135)	0.244 OR: 10.733 (0.198–580.643)	0.244 OR: 10.733 (0.198–580.643)
Autoimmune diseases (APS/SLE/SS/RA)	0 (0%)	3 (21.43%) 2 SLE and APS 1 SS	2 (28.57%) 1 SLE and APS 1 SS	1 (14.29%) 1 SLE and APS	0.012 OR: 49.000 (2.375–1011.10)	0.008 OR: 73.182 (3.114–1719.605)	0.032 OR: 37.154 (1.372–1006.271)
Other autoimmune diseases	0 (0%)	2 (14.29%) 1 AIT 2 CD	2 (28.57%) 1 AIT 2 CD	0 (0%)	0.028 OR: 32.200 (1.459–710.734)	0.008 OR: 73.182 (3.114–1719.605)	0.243 OR: 10.733 (0.198–580.643)
T1DM	0 (0%)	1 (7.14%)	1 (14.28%)	0 (0%)	0.082 OR: 17.889 (0.692–462.359)	0.032 OR: 37.154 (1.372–1006.271)	0.243 OR: 10.733 (0.198–580.643)
T2DM	0 (0%)	0 (0%)	0 (0%)	0 (0%)	0.396 OR: 5.552 (0.106–291.135)	0.244 OR: 10.733 (0.198–580.643)	0.244 OR: 10.733 (0.198–580.643)
Any kind of autoimmune disease (APS/SLE/SS/RA/T1DM/other)	0 (0%)	4 (28.57%) 1 SLE, APS, AIT 1 SLE, APS 1 SS, CD 1 T1DM, CD	3 (42.86%) 1 SLE, APS, AIT 1 SS, CD 1 T1DM, CD	1 (14.29%) 1 SLE, APS	0.006 OR: 69.000 (3.464–1374.494)	0.002 OR: 125.222 (5.576–2812.179)	0.032 OR: 37.154 (1.372–1006.271)
Trombophilic gene mutations	0 (0%)	2 (14.29%)	1 (14.29%)	1 (14.29%)	0.028 OR: 32.200 (1.459–710.734)	0.032 OR: 37.154 (1.372–1006.271)	0.032 OR: 37.154 (1.372–1006.271)
Parity							
Nulliparous	40 (50.0%)	10 (71.43%)	5 (71.43%)	5 (71.43%)	0.147 OR: 2.500 (0.724–8.636)	0.290 OR: 2.500 (0.458–13.649)	0.290 OR: 2.500 (0.458–13.649)
Parous—History of GH	0 (0%)	0 (0%)	0 (0%)	0 (0%)	0.286 OR: 9.000 (0.159–511.016)	0.185 OR: 16.200 (0.262–1000.106)	0.185 OR: 16.200 (0.262–1000.106)
Parous—History of HELLP and/or PE	0 (0%)	2 (50.0%)	1 (50.0%)	1 (50.0%)	0.009 OR: 81.000 (3.005–2183.284)	0.016 OR: 81.000 (2.232–2939.904)	0.016 OR: 81.000 (2.232–2939.904)
Parous—History of SGA/FGR	1 (1.25%)	0 (0%)	0 (0%)	0 (0%)	0.529 OR: 2.926 (0.103–83.058)	0.345 OR: 5.267 (0.168–165.318)	0.345 OR: 5.267 (0.168–165.318)
History of miscarriage (spontaneous pregnancy loss before 22 gestational weeks)	16 (20.0%)	1 (7.14%)	0 (0%)	1 (14.29%)	0.273 OR: 0.308 (0.037–2.529)	0.366 OR: 0.261 (0.014–4.800)	0.716 OR: 0.667 (0.075–5.938)
History of perinatal death (death of a fetus/newborn between 22 gestational weeks (or weighing over 500 g) and 7 days after the birth)	0 (0%)	0 (0%)	0 (0%)	0 (0%)	0.396 OR: 5.552 (0.106–291.135)	0.244 OR: 10.733 (0.198–580.643)	0.244 OR: 10.733 (0.198–580.643)
ART (IVF/ICSI/other)	2 (2.5%)	4 (28.57%)	1 (14.29%)	3 (42.86%)	0.003 OR: 15.600 (2.526–96.339)	0.149 OR: 6.500 (0.513–82.423)	0.001 OR: 29.250 (3.758–227.682)
Smoking during pregnancy	2 (2.5%)	0 (0%)	0 (0%)	0 (0%)	0.960 OR: 1.083 (0.049–23.740)	0.643 OR: 2.093 (0.092–47.761)	0.643 OR: 2.093 (0.092–47.761)
*Pregnancy details (first trimester of gestation)*							
Maternal age (years)	32 (25–42)	31 (24–49)	28 (24–35)	35 (26–49)	0.686	0.381	1.0
Advanced maternal age (≥35 years old at early stages of gestation)	18 (22.50%)	6 (42.86%)	2 (28.57%)	4 (57.14%)	0.115 OR: 2.583 (0.793–8.419)	0.715 OR: 1.378 (0.246–7.708)	0.060 OR: 4.593 (0.940–22.437)
BMI (kg/m^2^)	21.28 (17.16–29.76)	23.62 (15.67–31.53)	23.05 (15.67–31.53)	24.24 (18.97–27.43)	0.169	1.0	0.186
BMI ≥ 30 kg/m^2^	0 (0%)	1 (7.14%)	1 (14.29%)	0 (0%)	0.082 OR: 17.889 (0.692–462.359)	0.032 OR: 37.154 (1.372–1006.271)	0.244 OR: 10.733 (0.198–580.643)
Gestational age at sampling (weeks)	10.29 (9.57–13.71)	10.64 (9.14–11.71)	10.57 (9.71–11.57)	10.86 (9.14–11.71)	0.086	0.505	0.748
MAP (mmHg)	88.75 (67.67–103.83)	92.83 (78.33–108.25)	94.04 (78.33–108.25)	92.83 (83.67–107.17)	0.348	1.0	1.0
MAP (MoM)	1.05 (0.84–1.25)	1.09 (0.93–1.30)	1.13 (0.93–1.30)	1.09 (0.97–1.22)	0.440	1.0	1.0
Mean UtA-PI	1.39 (0.56–2.43)	1.32 (0.84–2.21)	1.32 (0.91–2.21)	1.56 (0.84–2.13)	0.952	1.0	1.0
Mean UtA-PI (MoM)	0.90 (0.37–1.55)	0.85 (0.56–1.38)	0.82 (0.56–1.38)	1.03 (0.57–1.31)	0.945	1.0	1.0
PIGF serum levels (pg/mL)	27.1 (8.1–137.0)	24.2 (14.6–32.6)	20.6 (14.6–32.6)	24.85 (18.2–29.2)	0.095	0.527	0.823
PIGF serum levels (MoM)	1.04 (0.38–2.61)	0.94 (0.65–1.18)	0.94 (0.69–1.18)	0.89 (0.65–1.15)	0.058	0.654	0.373
PAPP-A serum levels (IU/L)	1.49 (0.48–15.69)	1.50 (0.28–5.93)	1.16 (0.45–2.42)	1.68 (0.28–5.93)	0.668	0.375	1.0
PAPP-A serum levels (MoM)	1.17 (0.37–3.18)	0.93 (0.32–3.93)	0.69 (0.56–1.54)	0.95 (0.32–3.93)	0.258	0.168	1.0
Free b-hCG serum levels (μg/L)	60.21 (9.9–200.6)	54.34 (13.74–162.5)	45.35 (21.91–162.5)	55.56 (13.74–161.6)	0.364	0.649	1.0
Free b-hCG serum levels (MoM)	1.02 (0.31–3.57)	1.01 (0.28–2.55)	0.67 (0.47–2.55)	1.26 (0.28–2.47)	0.438	0.400	1.0
Positive screening for PE and/or FGR by FMF algorithm	0 (0%)	4 (28.57%)	2 (28.57%)	2 (28.57%)	0.006 OR: 69.000 (3.464–1374.494)	0.008 OR: 73.182 (3.114–1719.605)	0.008 OR: 73.182 (3.114–1719.605)
Aspirin intake during pregnancy	0 (0%)	5 (35.71%)	3 (42.86%)	2 (28.57%)	0.003 OR: 93.210 (4.772–1820.792)	0.002 OR: 125.222 (5.576–2812.179)	0.008 OR: 73.182 (3.114–1719.605)
*Pregnancy details (at delivery)*							
SBP (mmHg)	122 (100–155)	-	138 (102–157)	150 (134–210)	-	0.059	<0.001
DBP (mmHg)	76 (60–90)	-	80 (75–108)	98 (86–140)	-	0.189	<0.001
Proteinuria	0 (0%)	-	0 (0%)	7 (100%)	-	0.244 OR: 10.733 (0.198–580.643)	<0.001 OR: 2415.0 (44.642–130,644.585)
Gestational age at delivery (weeks)	40.07 (37.57–42.0)	35.79 (29.14–37.86)	36.86 (36.0–37.86)	32.71 (29.14–35.57)	<0.001	<0.001	<0.001
Delivery at gestational age < 37 weeks	0 (0%)	11 (78.57%)	4 (57.14%)	7 (100%)	<0.001 OR: 529.000 (25.636–10,915.759)	0.001 OR: 207.000 (9.217–4648.705)	<0.001 OR: 2415.000 (44.642–130,644.585)
RBC (×10^12^/L)	4.16 (3.49–4.85)	4.36 (3.43–4.81)	4.51 (3.93–4.81)	4.05 (3.43–4.38)	0.629	0.213	0.683
HGB (g/L)	119 (89–149)	121 (89–136)	122 (101–136)	120 (89–133)	0.565	0.800	1.0
TBIL (μmol/L)	0.251 (0.146–0.585)	0.672 (0.230–3.630)	0.637 (0.386–3.630)	0.950 (0.230–1.520)	<0.001	0.014	0.018
ALT (IU/L)	11.4 (5.40–18.61)	71.12 (12.05–531.81)	60.24 (12.05–435.77)	93.64 (21.01–531.81)	<0.001	0.004	<0.001
AST (IU/L)	19.21 (12.60–30.61)	128.75 (28.81–1128.45)	100.24 (28.81–1128.45)	151.86 (53.61–348.14)	<0.001	<0.001	<0.001
PLT (×10^9^/L)	237 (158–407)	100 (47–149)	104 (47–149)	100 (48–120)	<0.001	<0.001	<0.001
PT (s)	13.5 (11.1–15.8)	12.7 (11.1–14.1)	12.8 (11.1–14.1)	12.3 (11.1–13.5)	0.016	1.0	0.017
APTT (s)	34.3 (27–45.9)	33.8 (24.2–47.9)	36.9 (27.6–47.9)	30.8 (24.2–40.6)	0.377	1.0	0.371
FIB (mg/dL)	5.03 (3.99–7.38)	4.53 (1.48–6.80)	4.53 (2.75–6.80)	4.75 (1.48–5.91)	0.245	1.0	0.792
Fetal birth weight (grams)	3470 (2920–4240)	2410 (1230–3850)	2800 (2150–3850)	1845 (1230–2410)	<0.001	0.024	<0.001
Fetal sex							
Boy	40 (50.0%)	7 (50.0%)	4 (47.14%)	3 (42.86%)	1.000 OR: 1.000 (0.321–3.113)	0.718 OR: 1.333 (0.280–6.344)	0.718 OR: 0.750 (0.158–3.568)
Girl	40 (50.0%)	7 (50.0%)	3 (42.86%)	4 (57.14%)			
Mode of delivery							
Vaginal	69 (86.25%)	0 (0%)	0 (0%)	0 (0%)	<0.001 OR: 175.261 (9.765–3145.545)	0.003 OR: 90.652 (4.841–1697.654)	0.003 OR: 90.652 (4.841–1697.654)
CS	11 (13.75%)	14 (100%)	7 (100%)	7 (100%)			
Apgar score < 7, 5 min	0 (0%)	0 (0%)	0 (0%)	0 (0%)	0.396 OR: 5.552 (0.106–291.135)	0.244 OR: 10.733 (0.198–580.643)	0.244 OR: 10.733 (0.198–580.643)
Apgar score < 7, 10 min	0 (0%)	0 (0%)	0 (0%)	0 (0%)	0.396 OR: 5.552 (0.106–291.135)	0.244 OR: 10.733 (0.198–580.643)	0.244 OR: 10.733 (0.198–580.643)

Continuous variables, assessed using the Mann–Whitney or Kruskal–Wallis test, are presented as medians (ranges). Categorical variables, assessed using odds ratio testing, are presented as numbers (percents). *p*-values ^1, 2, 3^: the comparison between normal-term pregnancies and HELLP syndrome (hemolysis, elevated liver enzymes, and low platelets) and the comparisons between normal-term pregnancies and HELLP syndrome without PE or with PE, respectively. SLE, systemic lupus erythematosus; APS, antiphospholipid syndrome; SS, systemic scleroderma; RA, rheumatoid arthritis; AIT, autoimmune thyroid disease; CD, coeliac disease; T1DM, diabetes mellitus type 1; T2DM, diabetes mellitus type 2; GH, gestational hypertension; PE, pre-eclampsia; FGR, fetal growth restriction; SGA, small for gestational age fetuses; ART, assisted reproductive technology; IVF, in vitro fertilization; ICSI, intracytoplasmic sperm injection; BMI, body mass index; MAP, mean arterial pressure; UtA-PI, uterine artery pulsatility index; PIGF, placental growth factor; PAPP-A, pregnancy-associated plasma protein-A; b-hCG, beta-subunit of human chorionic gonadotropin; FMF, Fetal Medicine Foundation; SBP; systolic blood pressure; DBP, diastolic blood pressure; RBC, red blood cells; HGB, hemoglobin; TBIL, total bilirubin; ALT, alanine aminotransferase; AST, aspartate aminotransferase; PLT, platelets; PT, protrombin time; APTT, activated partial thromboplastin time; FIB, fibrinogen; CS, cesarean section.

## Data Availability

The data presented in this study are available on request from the corresponding author. The data are not publicly available due to rights reserved by the funding supporters.

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
