# Peer review of "First-Trimester Screening for HELLP Syndrome—Prediction Model Based on MicroRNA Biomarkers and Maternal Clinical Characteristics"

_ijms, 2023, doi:10.3390/ijms24065177_

Round 1

Reviewer 1 Report

General comments:

Unfortunately, there are some serious flaws in your manuscript. The most serious flaw is the usage of multivariate model, although only 14 HELLP syndrome is used in this model. If the number of main outcome is <20, multivariate model is not appropriate. The second flaw is that the authors did not validate this model. The created multivariate model by many variables used to show high sensitivity with low specificity. However, it is unknown, whether the model truly predict HELLP syndrome in another cohort. The third flaw is that this is just a case-control study. The prediction model should be created using a cohort study.

Comments:

Introduction: I agree that there are few predictive models for HELLP syndrome in the first trimester.

Materials and Methods:

Because you collected 4187 samples in the first trimester, you should construct prediction model using all samples.

The number of HELLP syndrome is too small to construct multivariable logistic regression model.

In such small number of primary outcome, the analyses should be restricted to univariate logistic regression, and show the relative risk of several risk factors on the development of HELLP syndrome.

The primary outcome should be restricted to complete (full) HELLP syndrome. Partial HELLP should be included in the category of HELLP syndrome.

Reviewer 2 Report

Article "First Trimester Screening for HELLP (Hemolysis, Elevated Liver Enzymes and Low Platelets) Syndrome – Prediction Model Based on MicroRNA Biomarkers and Maternal Clinical Characteristics" evaluated the potential role of microRNAs for early prediction of HELLP syndrome.

It presents new interesting information which  important for both science and practical medicine. There are no serious comments to the article. As a remark, in conclusion, it would be possible to discuss in more detail the algorithm for implementing this screening in the examination of pregnant women.

Reviewer 3 Report

The work is expertly done, the results are carefully analyzed, and the manuscript is well written. However, the paper would benefit from addressing the following points:

1.    Please, indicate that HELLP syndrome usually develops in III trimester of pregnancy, that is why the early predictive models are important.  

2.    A great part of Discussion repeats the data of statistical analysis already presented in Results section (lines 248-256, 275-296). Such overlaps can be eliminated without loss of quality.

3.    The manuscript is a part of a series of works focused on the miRNA expression in pregnancy-related complications of different origin and published by the same team recently. Taking this into account, it would be interesting if the authors systematize the findings obtained in this and previous studies and try to link each particular miRNA with specific pregnancy pathologies, for example, hypertensive or metabolic, at least in brief.

4.    Please, discuss the availability of miRNA screening in the clinical practice and indicate the advantage of such screening in comparison to routine maternal clinical characteristics only if the prediction model based on such characteristics yielded similar value (71 vs 78 % of pregnancies).

Reviewer 4 Report

This article showed that the combination of microRNA biomarkers and maternal clinical characteristics may be useful for the first trimester screening for HELLP syndrome. This is potentially interesting paper.

This paper seems to be generally written clearly, methods are sufficient and suitable to substantiate authors’ claims. Results are reasonable. The claims are fully supported by the experimental data and are not oversold. Discussion is based by previous literature and their results. But if this paper should be changed in some points, it would get better.

Page 9 or 10, Meterials and Methods: The definition of preeclampsia should be specified.

Figure 1, 2, 3, and 4: Abbreviations should be specified in each legend (e.g. ROC, CI, LR, PE, FGR, FPR, etc.).

Round 2

Reviewer 1 Report

Unfortunately, the revised version of manuscript was not changed according to my comments.